# Developing a Building Stock Model to Enable Clustered Renovation—The City of Leuven as Case Study

**Evelien Verellen** *\* and **Karen Allacker**

Department of Architecture, Faculty of Engineering Science, KU Leuven, 3001 Leuven, Belgium; karen.allacker@kuleuven.be
\* Correspondence: evelien.verellen@kuleuven.be

**Abstract:** The existing building patrimony is responsible for 36% of the global energy use and 37% of the greenhouse gas emissions. It is hence a major challenge to improve its energy performance. According to the Renovation Wave, the average annual renovation rate should be doubled by 2030 up to 3% and deep energy renovations should be encouraged. The Belgian city of Leuven works towards this target and is even more ambitious, setting their goal on becoming climate neutral by 2050. The strategy investigated in this study is to increase the renovation rate by clustering renovations, which is challenging since the Belgian building stock is highly privatised. Based on a thorough literature study, this paper examines various methodologies for building stock modelling. The main focus is comparing the required input data with the data availability, handling the data gaps, and defining their influence on the model's accuracy. The findings are applied to Leuven by analysing the main drivers to cluster renovation measures. However, many data gaps appeared, leading to the selection of a GIS-enhanced archetype model enriched by energy data as the most suitable approach. To avoid misinterpretation due to differences in data quality, transparent reporting in stock modelling is recommended.

**Keywords:** Renovation Wave; renovation rate; clustering; energy reduction; GIS; data; Leuven

## 1. Introduction

The building sector is currently responsible for 36% of the energy use and 37% of the greenhouse gas (GHG) emissions worldwide [1]. The European Union aims to lower the GHG emissions by 80 to 95% by 2050 compared to 1990 [2]. In line with the goals of the European Union, the Belgian government is working towards the same goal of evolving towards low carbon, sustainable, reliable, and affordable energy sources and towards a less energy-consuming society, and hence a better energy performing building stock. Considering that by 2050, two thirds of the world's population will live in cities, cities play a major role in achieving this objective [3]. This implies that cities not only have the opportunity, but also the responsibility to take the lead in reducing the energy needs of the building stock. Many cities already translated the roadmap of Europe into a long term climate action plan, such as the cities of Leuven, Ghent, Antwerp, and Mechelen in Flanders [4–7].

Over the past decade, an important transition in the energy performance of new buildings has been achieved in Europe and in Belgium thanks to the European Directive on the energy performance of buildings (recast) (2010) [8]. However, the existing building patrimony still remains an important challenge, as the energetic renovation of the existing stock is too slow to reach the GHG emission reduction goals. The current (2016) annual renovation rate in Belgium is less than 1%, while in order to reach the goals for 2050, this renovation rate should be increased to 3% [9].

Achieving a 3% renovation rate is a major challenge, especially because the majority of the buildings in Belgium are privately owned. This means that the renovation of each

building is based on the initiative of the building owner. By renovating building-by-building, the target of renovating 3% of the stock each year will most probably not be met, since one-by-one renovations will take more time and not all building owners are convinced yet or are reluctantly examining the administration work. A larger scale approach based on renovating clusters of buildings might be a valid way forward. The question arises as to how such clustered renovation could be accomplished.

The overall goal of this research, and of which this paper is a part, is to increase the renovation rate of a building stock in the Flemish context by clustering buildings with similar renovation needs and to identify the renovation measures with the highest GHG emission reduction potential at the stock level. The clustering of the buildings for renovation is based on the building envelope, the construction year, the building typology, and the ownership. The type of HVAC system and the airtightness of the buildings are not considered for clustering the buildings, as the main focus is on renovating the building envelope to investigate the GHG reduction potential using clustered partial or full envelope renovation. This methodology is fully elaborated using the case study of Leuven. This city is chosen for this study because of its engagement in its climate-neutral program by 2050, where the aim is to decrease the $CO_2$ emissions by 67% by 2030, and even by 80% by 2050 compared to the situation in 2010 [4]. In order to reach this goal and to evaluate the effect of the initiatives taken, the total $CO_2$ emissions of the city of Leuven have been monitored yearly since 2010 and the carbon footprint of the city has been calculated for the year 2010 [10]. In order to reach this target, the current renovation rate should be increased, and the goal has been set at 1000 renovations per year by 2030. Within the overall goal of this research, this paper focuses on the search for the most suitable building stock model that allows the identification of clustered renovation opportunities and enables the evaluation of the GHG impact reduction obtained by clustered renovation. The focus of this paper is defining the (data) needs and identifying the barriers in order to develop a building stock model based on a profound analysis of the state of the art. The search for the building stock modelling (BSM) approach and the handling of the data gaps for developing the stock model for Leuven are described in detail. A building stock model is aimed for that provides detailed insight in each building of the stock, such as the building's geometry and physical characteristics, energy performance, ownership, and users' profile, linked to the geographical location of each of the buildings.

Different approaches are possible to develop a building stock model. A review of various BSM approaches is provided, describing the goal of the models, the methodology used, and the data needed. As data availability is often a big challenge for BSM and influences the level of detail (LOD) and the accuracy of the model, this is seen as an important selection criterion in the choice of the most appropriate modelling approach. Based on the review, a modelling approach is selected for the goal of this study. Insights in the argumentation of the choices made may also be helpful for selecting the most appropriate building stock model approach for other goals.

The next section describes the requirements for the building stock model to enable the identification of clustered renovation opportunities (Section 2), followed by a discussion of the existing building stock models in relation to the identified features (Section 3). In Section 4, the implementation for the context of Leuven is elaborated. The data availability and data gaps are discussed and the consequences of these on the modelling approach are revealed. In addition, both the possibilities and the difficulties are explained. How previous studies overcame these data gaps and how such approaches can be applied to the case of Leuven are examined in Section 5.

## 2. Building Stock Model Features for Clustered Renovation

Multiple large-scale renovation projects have already taken place for neighbourhoods consisting of nearly identical buildings [11,12]. These projects proved that such clustered renovation reduces the costs and speeds up the process significantly, as the organisation can be outsourced for the whole neighbourhood [13]. This not only improves the efficiency,

but also unburdens the occupants. In order to identify large-scale renovation opportunities to a building stock that is very diverse, a building stock model is needed that allows the clustering of buildings that are sufficiently similar.

An important feature of the stock model to enable clustering is information on the geographical location of the buildings. If the distances between buildings are too large, clustering does not make sense. Hence, the stock model needs to include information on the location of each building and should therefore be Geographic Information System (GIS)-based. To allow for the identification of clustering opportunities beyond street level, a model of the whole city is also needed. Such a model enables the identification of multiple opportunities for clustering buildings, as this should not be solely limited to similar geometries.

Besides the location, the following primary data have been found to be required to categorise buildings with similar renovation needs: building geometry in at least LOD2 and preferable LOD3 (see Figure 1) [14], building typology, and renovation history, including the insulation level of each building element of the building envelope and build-ups of the building envelope (solid or timber frame construction).

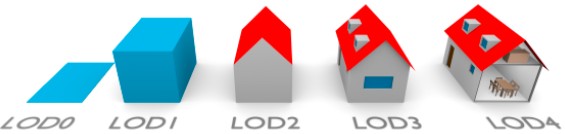

**Figure 1.** The five LODs of CityGML 2.0. Reprinted with permission from ref. [14]. Copyright 2016 Elsevier.

Besides the building-related data, information about the building owner is important, as ownership, age, and type of family might influence the willingness to renovate or influence the type of renovation and related available budget [15–17].

After selecting clusters of buildings with similar renovation needs, the details of the most-preferred renovation measures need to be developed based on the existing state of the buildings and building elements, and in view of reducing the life cycle of GHG emissions.

In the following section, different stock modelling approaches are analysed in view of the above features, and which of these approaches include the required features is determined. The most promising approaches are then further studied to select the one most appropriate for the goal of this study.

## 3. Building Stock Modelling Approaches

Literature on BSM approaches was searched using the keywords 'building stock', 'building stock model(l)ing' and 'building stock mode(l)ling city scale' on Google Scholar with publication dates between 2015 and 2021. Based on the papers found, additional literature was retrieved through the corresponding recommended articles in the references of the selected articles of the first step. The resulting articles were screened in four subsequent exclusion rounds: a full title analysis, an abstract analysis, a conclusion analysis, and a full paper analysis. Papers were selected based on the four following preconditions:

1. The main focus is bottom-up energy modelling or building parameter modelling;
2. The object of study is a residential building stock, so studies including only a single building or other building functions were excluded;
3. The building stock is located in a western country to provide a good basis for comparison with the case study (Europe, Oceania, and the United States of America);
4. Projects with multiple publications were only included once.

In total, 38 papers were analysed in depth. After the in-depth analysis, 25 papers remained relevant for the scope of this study and were reviewed in more detail. Their most relevant findings are summarised in Tables 1 and 2. Papers that did not meet the features required, but provided important insights on the topic, are included in the Discussion section.

### 3.1. Types of Stock Modelling Approaches

According to Kavgic et al. [18] and Mastrucci et al. [19], building stock models are divided into two different approaches: top-down and bottom-up. Top-down approaches are mostly used for the input–output modelling of cities at an aggregated level, and do not allow the identification of the main drivers of these inputs and outputs at a high granularity level. As the stock model in this research aims to identify buildings with similar renovation needs, a top-down approach is not suitable.

Bottom-up approaches start from buildings or building components at a disaggregated level, and combine these to the building stock scale [18]. As indicated in Table 2, bottom-up models allow the tracing of all contributions to the level of the stock components and form a robust basis for future scenario modelling [19], which makes a bottom-up approach the preferred approach for the goal of this study. However, bottom-up models require a larger amount of input data than top-down models, and thus, many studies highlight the (lack of) availability of the required input data for the full building stock as one of the biggest challenges. Various approaches can be used to develop a bottom-up building stock model. In order to define the most appropriate approach for the goal of this study, it is important to understand the features, limitations, and data needs of the various methods. These insights are gained through a literature study of existing bottom-up models and are summarised in Table 2.

Most studies indicate that a higher LOD of the bottom-up approach leads to a higher accuracy of the model [18,19]. However, Willmann et al. [20] contradict this, explained by the fact that the LOD and the quality of the input data are the determining factors for the accuracy. Hence, the trustworthiness of the input data is crucial, and as in their study the database could not be calibrated with measured energy use data, the higher LOD of the bottom-up model did not result in a higher accuracy.

### 3.2. Bottom-Up Stock Modelling Approaches

Different bottom-up approaches have been identified in the literature, as summarised in Table 1. A first approach is the building-by-building approach [21]. In this approach, the data of interest need to be known for each single building in the stock, or one should be able to calculate the data for each building in the stock. This method results in a highly accurate building stock model with a high level of granularity. This type of model, for example, allows the identification of the main reasons for the high energy use of each individual building and, hence, the renovation potential. The major drawback of this method is the need for big data [21].

**Table 1.** Overview of bottom-up building BSM approaches and their data needs [21–45]. The results of the different studies are summed, all characteristics that occur in more than 60% of the papers are indicated in green, and all characteristics that occur in more than 40% of the papers are indicated in orange.

| Literature Source | Country | Method | Goal | Construction Period/Year | Year of Refurbishment | Typology | Roof Type | GIS-Dataset | Measured Energy Consumption | Calculated Energy Consumption | Measurement Data by Remote Sensing | Building Footprint | Building Area | Building Volume | Number of Stories | Height | Building Geometry | U-Values | Material Use | W2W | Occupancy | HDD/CDD | EPC Data | Archetype Buildings | Land Use | Heating Installation Data |
|---|---|---|---|---|---|---|---|---|---|---|---|---|---|---|---|---|---|---|---|---|---|---|---|---|---|---|
| Breunig et al. [40] | United States | top-down + bottom-up, archetypes | a broadly applicable approach for modeling future commercial, residential, and industrial floorspace, thermal consumption (heating and cooling), and associated GHG emissions at the tax assessor land parcel level. | yes | no | yes | no | yes | no | no | no | yes | no | no | no | no | no | no | no | no | no | no | no | yes | no | no |
| Brogger et al. [33] | Denmark | bottom-up, building-by-building | a hybrid bottom-up building stock energy model was developed in order to overcome the drawbacks of traditional building-physics (engineering) based modelling methods. Using a sample of more than 100,000 residential buildings, individual building-physics based models were calibrated against energy use data in a multiple linear regression setting, thereby providing a novel hybrid bottom-up building stock energy model. | yes | no | yes | no | no | yes | yes | no | no | yes | yes | no | no | no | no | no | no | no | hdd | yes | no | no | no |
| Buffat et al. [29] | Switzerland | bottom-up, building-by-building | model building heat demand, derive envelope of all buildings | yes | no | no | yes | yes | no | no | no | yes | no | no | yes | yes | no | no | no | no | yes | no | no | no | no | yes |
| Caputo et al. [46] | Italy | bottom-up, archetypes | hybrid approach: real + statistical data identification of main renovation barriers and drivers | yes | yes | yes | no | yes | no | yes | no | no | yes | yes | no | yes | yes | no | no | yes | no | no | yes | yes | no | no |
| Dall'o et al. [27] | Italy | bottom-up | Using a GIS platform the integration of two data sources (sample buildings) | yes | no | yes | yes | yes | no | yes | no | yes | yes | yes | yes | yes | yes | yes | no | yes | no | yes | no | no | no | no |
| D'Alonzo et al. [32] | Italy | bottom-up, building-by-building | The paper presents a spatially explicit and "bottom-up" methodology for the building stock analysis of the residential sector. the energy balance at the building level (BL) for the whole Valle d'Aosta region (Italy) is addressed | yes | no | no | no | yes | no | yes | no | yes | yes | yes | yes | yes | yes | no | no | no | no | hdd | yes | no | yes | yes |
| Evans et al. [34] | United Kingdom | bottom-up building-by-building | the development of a new three-dimensional model of the British building stock, called '3DStock'. | yes | no | yes | yes | yes | part | no | no | yes | yes | yes | yes | yes | yes | no | yes | yes | yes | no | yes | no | no | yes |
| Garcia (2018) [21] | Spain | building-by-building,+ archetype | LCA of renovation measures | yes | no | yes | no | yes | no | no | no | yes | yes | no | yes | yes | yes | no | no | no | no | no | no | yes | no | no |
| Gendebien et al. [22] | Belgium | bottom-up, archetypes | model and simulate domestic energy use | yes | no | yes | no | no | no | no | no | no | yes | no | no | no | no | no | no | no | no | hdd | no | yes | no | yes |

**Table 1.** *Cont.*

| Literature Source | Country | Method | Goal | Construction Period/Year | Year of Refurbishment | Typology | Roof Type | GIS-Dataset | Measured Energy Consumption | Calculated Energy Consumption | Measurement Data by Remote Sensing | Building Footprint | Building Area | Building Volume | Number of Stories | Height | Building Geometry | U-Values | Material Use | W2W | Occupancy | HDD/CDD | EPC Data | Archetype Buildings | Land Use | Heating Installation Data |
|---|---|---|---|---|---|---|---|---|---|---|---|---|---|---|---|---|---|---|---|---|---|---|---|---|---|---|
| Gulotta et al. [23] | Europe | bottom-up, archetypes | bottom-up modelling, energy dynamic simulation and LCA to evaluate different renovation strategies | yes | no | yes | no | no | no | no | no | no | no | no | no | no | no | no | no | no | no | yes | no | yes | no | yes |
| Kontokosta et al. [35] | United States | bottom-up, building-by-building | develop a predictive mode of energy use | yes | no | no | no | no | part | no | no | yes | yes | yes | yes | yes | no | no | no | no | yes | no | no | no | yes | no |
| Nageli et al. [36] | Switzerland | bottom-up | BSM based on agent-based modeling approach by modeling indiviual decisions on building level. | yes | yes | yes | yes | no | no | yes | no | yes | yes | yes | yes | yes | yes | yes | no | yes | no | no | no | no | no | yes |
| Nishimwe et al. [37] | Belgium | bottom-up, building-by-building | the annual heat consumption and heat demand of Wallonia building stock of more than 1,700,000 buildings are assessed | no | no | yes | no | yes | no | no | no | no | yes | no | yes | no | no | no | no | no | no | hdd | no | no | no | no |
| Nutkiewicsz et al. [24] | United States | bottom-up, archetypes | assess the feasibility in using an integrated data-driven urban energy simulation model to quickly evaluate various large-scale retrofits in an urban environment | yes | no | yes | yes | yes | no | no | no | yes | no | no | yes | yes | yes | no | yes | no | no | no | no | yes | no | no |
| Österbring et al. [25] | Sweden | bottom-up, archetype | describe urban building stock | yes | yes | yes | no | yes | part | no | no | yes | yes | 2.5d | yes | yes | no | no | no | no | no | no | yes | yes | no | yes |
| Pittam et al. [26] | Ireland | bottom-up, archetypes | develop archetype buildings | no | no | no | no | yes | no | no | no | no | no | no | no | no | no | no | no | no | no | no | no | no | yes | no |
| Pittam et al. [30] | Ireland | remote sensing | understanding the stock | no | no | no | yes | no | no | no | yes | yes | yes | yes | yes | yes | yes | no | no | yes | no | no | no | no | no | no |
| Schiefelbein et al. [40] | Germany | bottom-up, archetypes | urban energy modeling approach based on open source GIS dataset to reduce input data uncertainty and simplify city district modeling | yes | no | yes | no | yes | no | no | no | yes | yes | no | yes | yes | no | no | no | no | no | hdd | no | yes | no | no |
| Stephan et al. [41] | Australia | bottom-up, archetypes | spatially model building stock and quantify embodied environmental requirements | yes | no | yes | yes | yes | no | no | no | yes | yes | no | yes | yes | yes | yes | yes | yes | no | no | no | yes | yes | no |
| Streicher et al. [42] | Switzerland | bottom-up, archetypes | simulate national building energy demand + renovation scenarios | yes | no | yes | no | no | no | no | no | no | yes | no | no | no | yes | yes | no | yes | no | yes | no | yes | no | yes |
| Taylor et al. [38] | United Kingdom | bottom-up, building-by-building | urban scale energy modelling | no | no | no | no | yes | no | no | no | yes | no | 2.5d | no | yes | 2.5d | no | no | no | no | no | no | no | no | no |

**Table 1.** *Cont.*

| Literature Source | Country | Method | Goal | INPUT DATA (Per Building) | | | | | | | | | | | | | | | | | | | | | | |
|---|---|---|---|---|---|---|---|---|---|---|---|---|---|---|---|---|---|---|---|---|---|---|---|---|---|---|
| | | | | Construction Period/Year | Year of Refurbishment | Typology | Roof Type | GIS-Dataset | Measured Energy Consumption | Calculated Energy Consumption | Measurement Data by Remote Sensing | Building Footprint | Building Area | Building Volume | Number of Stories | Height | Building Geometry | U-Values | Material Use | W2W | Occupancy | HDD/CDD | EPC Data | Archetype Buildings | Land Use | Heating Installation Data |
| Usman Ali et al. [44] | Ireland | bottom-up, archetypes | developing a methoddology based on bottom-up approach for GIS based residential building energy modeling at district scale > greatest potential for energy savings | yes | no | yes | no | yes | no | yes | no | yes | no | yes | yes | yes | yes | no | no | yes | yes | no | yes | yes | no | yes |
| van der Bent et al. [43] | Netherlands | bottom-up, archetypes | to investigate the extent to which empirical models provide more accurate estimations of actual energy consumption when compared to a theoretical building energy model, in order to estimate average actual energy savings of renovations. | yes | no | yes | no | no | yes | yes | no | yes | yes | no | no | no | yes | no | no | yes | no | no | no | yes | no | yes |
| Wang et al. [28] | Switserland | bottom-up, archetypes | energy demand model, retrofitting model | yes | no | yes | yes | yes | no | no | no | yes | no | no | no | yes | 2.5d | no | no | yes | yes | no | no | yes | no | yes |
| Wurm et al. [39] | Germany | bottom-up, remote sensing | a workflow for deep learning-based building stock modeling using aerial images at a city scale for heat demand modeling + evaluating renovation scenarios | yes | no | yes | no | yes | no | no | yes | yes | no | no | no | yes | no | no | no | no | no | no | no | no | no | no |
| | | | Yes | 21 | 3 | 19 | 7 | 17 | 5 | 7 | 2 | 18 | 16 | 11 | 14 | 17 | 14 | 4 | 2 | 11 | 5 | 8 | 6 | 14 | 3 | 11 |
| | | | No | 4 | 22 | 6 | 18 | 8 | 20 | 18 | 23 | 7 | 9 | 14 | 11 | 8 | 11 | 21 | 23 | 14 | 20 | 17 | 19 | 11 | 22 | 14 |
| | | | Total | 25 | 25 | 25 | 25 | 25 | 25 | 25 | 25 | 25 | 25 | 25 | 25 | 25 | 25 | 25 | 25 | 25 | 25 | 25 | 25 | 25 | 25 | 25 |

**Table 2.** Strengths and limitations of different BSM approaches [18–45,47,48].

| Method | Approach | Strengths | Limitations | Literature Source |
|---|---|---|---|---|
| Top-down | General | - input–output modelling of cities<br>- easy scenario modelling at large scale | - not possible to identify main drivers of energy use (GHG emissions) | Breunig et al. [40], Kavgic et al. [18] |
| Bottom-up / Building-by-building approach | General | - provides a higher resolution as particular effects can be traced back to a construction element/material/building<br>- interesting for both building owners and policy makers<br>- robust basis for future scenario modelling | - inconsistencies in available urban energy data<br>- available building data too limited<br>- absent/limited/static occupant profiles<br>- size of the database vs size of the stock<br>- influence of the reference unit<br>- need for greater data transparency and data access from utility providers<br>- flexibility of the model for different contexts | Dall'O et al. [25], Kontokosta et al. [33], Stephan et al. [32], Streicher et al. [38], Taylor et al. [36], Usman Ali et al. [48], Wurm et al. [37] |
| Bottom-up / Building-by-building approach | General | - high accuracy<br>- high level of granularity<br>- easy to identify high energy consumers | - necessity for detailed data<br>- possible higher computational time<br>- user behaviour variations<br>- -condition of the building elements often not included (quality, maintenance level, etc.) | Buffat et al. [47], Garcia et al. [21], Mastrucci et al. [19], Österbring et al. [23] |
| Bottom-up / Building-by-building approach | Prediction model at building level based on top-down data using machine learning algorithms | - building-by-building data are scaled up to the urban level<br>- use of GIS/data driven predictive model<br>- possible to assess indoor conditions<br>- fills data gaps of building parameters, occupancy, or energy data<br>- higher accuracy than theoretical building energy model | - purely data-driven approaches lack underlying thermodynamic modelling to ascertain how energy retrofits might affect future energy performance<br>- availability/quality of data<br>- accuracy for each part of the study may differ (e.g., each renovation measure) | Gao et al. [45], Kavgic et al. [18], Kontokosta et al. [33], Nutkiewicsz et al. [43], Schiefelbein et al. [22], van der Bent et al. [39] |
| Bottom-up / Building-by-building approach | Model based on energy calculations | | - performance gap (prebound and rebound effect)<br>- accuracy based on the reliability of input data<br>- strongly dependent on assumptions made | Gendebien et al. [41], Österbring et al. [23], van der Bent et al. [39] |

**Table 2.** *Cont.*

| Method | Approach | Strengths | Limitations | Literature Source |
|---|---|---|---|---|
| | Archetype approach | - less data needed (link to archetype based on known parameters) | - energy demand of old single-family houses overestimated (renovations unknown, unheated spaces, etc.) and of new MFH underestimated<br>- uncertainty regarding social technical drivers of energy use<br>- data to link buildings to archetypes required<br>- number and representativeness of archetypes | Buffat et al. [47], Kavgic et al. [18], Pittam et al. [27], Wang et al. [26], Willmann et al. [20] |
| | GIS-enhanced archetype classification | - limited data needed<br>- building specific data<br>- identification of particular buildings, their quantification and distribution | - accuracy of the building characteristics is limited to the archetype definition | Caputo et al. [46], Lismont et al. [44] |

A second type is the archetype approach [20–23,27,32,38–43,46,48,49]. In this approach, the buildings are represented by archetype buildings based on a classification system. The most common classification system is the age-type classification to approximate the U values of the buildings [25,43,44], but there are also other classification approaches: age–size–type classification to approximate the materials of the buildings for LCAs [21], classification based on age, type, level of urbanisation, fuel type, and building elements [43], classification based on age, type, function, and land use [42], classification based on age, type, and geometry [22,46], age–type–geographic location classification [42], type–location classification [40], and finally age–type–insulation–installations classification [41]. The number of archetypes and hence the amount of data to be collected differs widely depending on the classification system, but is less than for a building-by-building approach. An interesting source of information regarding archetypes for residential buildings in Europe is the TABULA/EPISCOPE project, in which the age–type classification is used [50]. A combination of the first two bottom-up modelling approaches is sometimes used as well. García-Pérez et al. [21], for example, use a building-by-building model using GIS to assess thermal upgrades of buildings in the metropolitan area of Barcelona, combined with archetypes to fill the data gaps.

A third bottom-up approach identified in this literature study is a GIS-enhanced archetype model [44]. In such a model, the archetype approach is used, but supplemented by GIS data in order to add some building-specific data to the stock model. Examples of building parameters replacing the generic data of the archetype buildings are the measured building geometry and address details, and in case of local extensive datasets, even more parameters can be replaced. This approach is interesting in case a building-by-building approach is preferred, but not all features are known for each building in the stock, or in case extra building specific information is useful in an archetype approach. By including GIS data in the general archetype information, the model becomes more accurate and reliable [21–23,25–27,32,40,43,46–48]. These GIS data can be supplemented using building heights or using LiDAR data to create a 2.5D or even 3D city model [51]. In the study of Mastrucci et al. [52], a GIS-enhanced model is used, but in this case, the data gaps are filled using statistical data that are apportioned to different building types instead of using predefined archetypes. In the study of Schiefelbein et al. [22], the GIS model is enhanced by filling missing values based on national statistics.

A fourth and final modelling approach consists of upscaling a limited number of building-by-building data to the full stock, making use of GIS data. For example, Kavgic et al. [18] describe the Energy and Environmental Prediction model, in which geolocated building-by-building data are processed individually and then scaled up to the urban level to assess the impact of indoor conditions on human health. As this model starts from a limited number of buildings including all needed building parameters (bottom-up data) and upscales to the full building stock (top-down data), this approach is not useful for the scope of this study as the granularity of the data is reduced.

Table 2 illustrates the other benefits of machine learning-based prediction models. For instance, these can be used to fill the data gaps of building parameters, occupancy, or energy use. The study of van der Bent et al. [39] even reveals that prediction models using machine learning algorithms to define building parameters and energy use might achieve a higher accuracy than theoretical building energy models using energy simulations. However, theoretical models are necessary to simulate future scenarios.

### 3.3. Data Availability

In Table 1, the different bottom-up modelling approaches and the necessary associated input data are visualised. Out of the 25 analysed papers describing a bottom-up BSM methodology, 13 papers use an archetype approach, 7 a building-by-building approach, and 1 a combination of both. Two papers make use of representative building agents, which are more disaggregated and more adjustable when linking to the stock than using the archetype approach. In addition, two papers use a remote sensing approach to inventory

the geometric characteristics of the stock that needs additional data to estimate the energy use. In the study of Pittam et al. [24], a geometry building stock model (LOD3) is developed by a remote measurement and mapping technique using Google Earth. Table 1 shows that the construction period, building typology, GIS dataset, building footprint, building area, number of stories, and building height are the most common building input parameters. Each of the parameters occurs at least 14 times. These parameters are followed by the building volume, the complete building geometry, the window to wall ratio, the heating installation, and the presence of archetype buildings, which all occur at least 11 times. U values, measured energy use, or calculated energy use are only known in a very limited number of studies, so most studies need to derive, calculate, or predict the energy use of the buildings. The occupancy of the buildings is only known in five studies (improving the accuracy of the energy calculations), and the material use is only known in two studies. It can be concluded from this analysis that the methods used in most of these studies do not allow for an accurate estimation of the renovation needs of the building stock, since no detailed input parameters regarding the energy performance of the building envelope or efficiency of the technical installations are known. If the data gaps in these studies could be resolved, a more detailed estimation of the renovation potential of the building stock would be possible. Methods to fill the data gaps are discussed in Section 3.6.

### 3.4. Energy Performance of Building Stock

Four data sources are commonly used to define the energy-related characteristics of a building stock: (1) measured data and (2) statistical data, which can both be at the building level or at a higher scale (city, statistical sector, street); and (3) steady state or (4) dynamic building energy simulations [22,26]. The selection of the most appropriate data source is based on the availability of the data and the goal of the stock model. On the one hand, measured energy use per building is the most accurate method to identify high energy-consuming buildings (e.g., [53]). On the other hand, building energy simulations are required to assess the effect of renovation measures. In this case, the calculation method is selected based on the amount of available data and the need for precision [23,47]. The result is highly dependent on the assumptions made and the data quality. When using building energy simulations, it is recommended to consider the energy performance gap, i.e., prebound and rebound effect, to avoid steering in the wrong direction [22,25,43]. Various data sources can also be combined: in the research of Buffet et al. [47], the measured energy use is known for 1845 buildings in the city of St. Gallen, Switzerland, and this information is used to validate the energy calculations. In the study, iterations of 2000 heat profiles for each building were simulated and a Monte Carlo analysis was performed. However, one should be cautious about the reliability of the results, as the energy demand of old single family houses is consistently overestimated, while the energy demand of new multifamily houses is often underestimated [47].

### 3.5. Material Inventory of Building Stock

A building stock model can also be used to estimate the type and amount of materials present in the stock and assess the related embodied environmental impacts. For example, García-Pérez et al. [21] perform an environmental assessment at the urban level using a bottom-up methodology combining GIS and Life Cycle Assessment (LCA) methods. In the study of Stephan and Athanassiadis [32], a spatially defined bottom-up model is developed using building archetypes to represent the land use, construction period, and building height for Melbourne, Australia. Using these models, the material flows of the building stock can be analysed in order to quantify the inputs, the outputs, and the materials stored in the building stock of cities and to assess the corresponding environmental impacts. In the study of Heeren et al. [54], based on an in-depth building-by-building dataset, the environmental impact of the building stock is simulated using a life cycle approach for both the used energy and the materials by clustering the buildings according to their building type, construction period and heating, ventilation, and air conditioning (HVAC) system.

### 3.6. Data Enrichment Methods

Besides the four data sources, methods can be used to enrich the data from these four sources in case of data gaps for specific buildings in the stock. Different methodologies to fill the data gaps are found in the literature (the evaluated papers presented in Table 1). Generally, the archetype approach can be seen as a way to avoid the need for building-by-building data, as only a limited number of building parameters are required in such an approach to link each building to the corresponding reference building and then approximate all buildings by the detailed information of the archetypes [22,23,25,30,40].

The first enrichment approach is to approximate the missing data for some buildings by assuming the data are identical to the known data of neighbouring buildings. For this enrichment method, GIS data are required. This approach is used in the study of Schiefelbein et al. [22], where some missing values were filled using an **enrichment approach using GIS**. More specifically, for each missing construction year of a building in the database, GIS tools are used to examine the neighbourhood of 10.000 m$^2$ to predict the most likely construction year.

A second approach is based on applying **machine learning algorithms** to complete datasets based on similar buildings [18,24,35,40,43,48]. In this case, the missing data should be available for a subsection of the building stock so that the prediction model can use this subsection as a training dataset. For example, a subset including measured energy use of buildings is then used to predict the energy use of the buildings in the stock for which data are lacking [18,24,35,43,48,55]. On the other hand, the energy use of buildings in the stock can also be estimated based on energy simulations using building features, which are predicted with machine learning algorithms. Different machine learning algorithms were identified in the literature study (Table 1) to predict missing data. In the literature, clustering techniques are applied in order to find similar buildings in an unbiased way [33,45]. When a similar building is found based on a few parameters, the other parameters are assumed to be similar as well. One frequently used clustering algorithm is K-means clustering. In the study of Gao and Malkawi [45], the GIS database is enriched by distributing the buildings into different representative building categories using the K-means clustering method. The centroids of each of the clusters are the most-suited representative buildings, and these are used in this study to define the benchmark of this building stock. These benchmarks are used to assess the energy performance of similar buildings. Another machine learning approach used in the literature is the multiple linear regression model for downscaling. In the study of Mastrucci et al. [52], statistical data are used to feed the model and determine the energy use at the dwelling level. For each building, a prediction is made per archetype, per floor area, and per person. The algorithm starts at the postcode level and downscales both the natural gas and electricity use to the individual building level based on different building parameters. Kontokosta and Tull [33] estimate the energy use of 1.1 million buildings in New York City based on the available information of 23.000 buildings using a predictive model that utilises a machine learning approach. More specifically, based on the energy benchmark data of the city, the electricity and gas use of every building is predicted using linear regression, random forest, and support vector regression algorithms. Finally, Artificial Neural Networks (ANN) can be used to enrich the data in the building stock model. This is used in the study of Wurm et al. [37], for example, to extract the building form from aerial images in order to fill the geometry data gaps. In the comparative study of Seyedzadeh [55], both ANN and support vector machines (SVM) are revealed to be the most common algorithms to predict the heating load and the energy demand of buildings.

A third approach often used to fill data gaps is **allocating top-down data** to buildings [40,56]. The top-down data can be based on statistical information, literature, or surveys. Based on a probability distribution, values for building parameters are assigned to the buildings in the stock. In this case, the relevant building parameters are not known at the building level, but only at the stock, statistical sector, or national level, and this distribution is further extended to the buildings of the stock by randomly assigning values according to this distribution.

By filling in the data gaps in stock modelling, the accuracy of the model reduces and the uncertainty increases. Hence, it is important to validate the model, e.g., by comparing to results from top-down models, and to gain insight in the uncertainty of the model. The latter is further discussed in Section 5.2.

### 3.7. Uncertainty Assessment

The need for big data in stock modelling typically requires combining various databases, and filling data gaps for at least part of the stock. The reliability of the data from these different sources, which might even use different system boundaries, has a significant influence on the uncertainty of the results. Documenting the assumptions is, hence, an important requirement for transparently reporting these uncertainties.

In the studies of Österbring et al. [23] and Breunig et al. [40], uncertainties are assessed using sensitivity analysis for the most impactful uncertain parameters. In addition, in the study of Garcia et al. [21], the influence of the geometry assumptions was verified using a sensitivity analysis, revealing a link between the average height and the window-to-wall ratio, which reduces the uncertainty of allocating impacts to a specific geolocation. Similarly, in the study of Schiefelbein et al. [22], multiple simulations are run for the uncertain parameters, with 10,000 samples per parameter being assumed. Based on 10,000 simulations, a probability density function was created to identify the most likely values. The resulting normalised net space heating demand is revealed to be close to a gaussian distribution. Kavgic et al. [18] and Garcia et al. [21] emphasise that the relative importance of the input parameter variations on the predicted output need to be quantified.

Buffat et al. [29] use a Monte Carlo analysis to assess the effect of the uncertainty of input parameters on the model outcomes. More specifically, to minimise uncertainties in the simulated heat demand, a normal distribution of geometric input values and indoor temperatures is taken, a uniform distribution for the shadow factor, a triangular distribution for the thermal storage capacity, and a lognormal distribution for the ventilation rates. Then, a Monte Carlo simulation with 2000 iterations is performed.

In other studies, the uncertainty of the model is reduced by checking the past development of the building stock with the statistics from past years [36,57]. Using both historic and current data for the building stock, the simulated long-term development in the building stock can be validated. This validation step also guarantees better simulations for future scenarios [36,57]. D'Alonzo et al. [32] validated the results of the bottom-up model by comparing these with the available top-down data.

A final uncertainty that should be taken into account is the discrepancy between the estimated and measured energy use of buildings, which can be more than 100% [24,29]. In the research of Kavgic et al., a lack of knowledge regarding the real energy use of residential buildings is clarified by the fact that energy models cannot handle social interactions properly. The uncertainty regarding the occupants' behaviour is addressed as a risk of unpredictability in the studies, but it is not resolved.

### 3.8. Conclusions

In the overall goal to develop a stock model that allows the identification of buildings with poor energy performance and a high environmental impact, their location, and the main drivers of the energy use and impacts, a literature study was performed of existing modelling approaches. From the review, it can be concluded that top-down models are not sufficiently detailed for this purpose, and only bottom-up models are appropriate. Nevertheless, a top-down model can still be used to validate the bottom-up model results.

As the location of the buildings is important to identify clusters for group renovation, the model should include geographical information (GIS data). A building-by-building approach is preferred in terms of accuracy of the building parameters, energy related data, and occupant characteristics, but an archetype-based approach might serve as alternative in cases where data are not sufficiently available for each single building. Furthermore, the dataset can also be extended using different enrichment approaches, as discussed in

Section 3.6. A mixed approach might be a solution to reach a high level of accuracy if data are incomplete.

## 4. Case Study—Leuven

This study focuses on the residential building stock of Leuven. The stock of Leuven in the year 2021 consists of 34,206 buildings and 64,283 residential units [57]. The building stock is very diverse and highly privatised [58]. As most buildings are privately owned, large-scale renovation projects are not only challenging in terms of construction, but also in terms of engaging the building owners. Although this paper focuses on modelling the stock of Leuven, the approach can be extrapolated to other cities with a similar climate, i.e., requiring space heating and no mechanical cooling. However, an analogue methodology can be developed including or replacing more building parameters when developing a stock model for a different climate.

### 4.1. Availability of Required Data

A first important source for developing the stock model of Leuven is the open-source GIS data ('GRB Flanders') [59] of the Flemish region. This database includes various data of the buildings: area of the ground floor of the building, perimeter, postal address, and ridge height (LOD1), as indicated in Table 3. However, this LOD is insufficient for the goal of this study.

**Table 3.** Ten samples from the dataset of Flanders (anonymised by removing the house address numbers).

| | | | | | | | | | |
|---|---|---|---|---|---|---|---|---|---|
| **Flanders Dataset** | | | | | | | | | |
| **NR** | **ID_Flanders** | **Street** | **Municipality** | **NR** | **Height** | **Perimeter** | **Footprint** | **Centroid_X** | **Centroid_Y** |
| 1 | 2880957 | Oude Rondelaan | Leuven | XX | 9.86 | 41.07 | 104.11 | 172,655.45 | 175,468.68 |
| 2 | 2880955 | Oude Rondelaan | Leuven | XX | 7.89 | 50.28 | 154.28 | 172,683.67 | 175,484.70 |
| 3 | 2880944 | 's Hertogenlaan | Leuven | XX | 6.3 | 71.22 | 222.35 | 172,565.03 | 175,460.76 |
| 4 | 2880943 | 's Hertogenlaan | Leuven | XX | 6.1 | 54.06 | 164.48 | 172,549.18 | 175,450.56 |
| 5 | 2880937 | 's Hertogenlaan | Leuven | XX | 6.7 | 47.7 | 130.86 | 172,528.38 | 175,437.81 |
| 6 | 2885460 | Rotspoelstraat | Heverlee | XX | 7.49 | 54.46 | 137.34 | 170,646.96 | 172,205.77 |
| 7 | 2880722 | Brouwersstraat | Leuven | XX | 10.98 | 22.58 | 31.66 | 172,677.58 | 174,831.51 |
| 8 | 2880734 | Brouwersstraat | Leuven | XX | 10.93 | 38.9 | 66.33 | 172,613.70 | 174,869.47 |
| 9 | 2886057 | Parkbosstraat | Heverlee | XX | 13.5 | 54.16 | 163.06 | 173,147.07 | 172,260.91 |
| 10 | 2908196 | Politieke-Gevangenenlaan | Wilsele | XX | 9.01 | 40.87 | 71.56 | 174,227.56 | 178,078.37 |

In the case of Leuven, an additional GIS database is available, including all buildings of the stock and providing information regarding the building function, the number of floors, the construction year, the roof type, and the number of residential units, as shown in Table 4 [60]. Both databases are of interest, as they include different building information, and thus, combining both adds value to this research. Both databases work with different IDs and different polylines. In a previous step in this research, the majority of the buildings could be linked in both databases using geoprocessing tools [61]. This resulted in a GIS database for the building stock of Leuven with LOD2 in which all Flemish main building IDs (excluding annexe buildings) could be linked to a main building of the GIS database of Leuven. To illustrate the outcome, 10 corresponding samples from both databases are visualised in Tables 3 and 4. Both tables follow the same order and could thus be linked.

As the GIS databases do not include information on the energy performance or insulation level of the buildings, as illustrated in Tables 3 and 4, the data need to be further enriched based on different data sources. A potential source for this enrichment is the Energy Performance Certificate (EPC) database for Flanders. The EPC database provides information about the energy performance parameters of the buildings; more specifically, the following relevant data can be retrieved from the database: construction year, building typology, ownership, roof type, calculated energy use, U values, areas of building elements, EPC label of the building, and fabrication year of the HVAC installations. However, this

database is only available in an anonymised way for privacy reasons. The location (address) of the buildings is not provided, which means that the EPC database cannot be directly linked with the GIS databases of Flanders and Leuven.

**Table 4.** Ten samples from the dataset of Leuven (SFH: single family house, EDU: education).

| Leuven Dataset | | | | | | | | | |
|---|---|---|---|---|---|---|---|---|---|
| NR | ID_Leuven | Construction Year | Number Floors | Number Floors Basement | Number Floors Roof | Roof Type | Function | Centroid_X | Centroid_Y |
| 1 | 2264224 | 1966 | 2 | 0 | 1 | pitched | SFH | 172,655.4456 | 175,468.7454 |
| 2 | 2264232 | 1971 | 2 | 1 | 0 | pitched | SFH | 172,683.3761 | 175,485.0469 |
| 3 | 2264244 | 1976 | 2 | 0 | 1 | flat | SFH | 172,565.0013 | 175,460.8042 |
| 4 | 2264248 | 1988 | 1 | 1 | 0 | pitched | SFH | 172,549.1693 | 175,450.9007 |
| 5 | 2264252 | 1990 | 2 | 0 | 1 | flat | EDU | 172,528.3674 | 175,437.8730 |
| 6 | 2407336 | 1988 | 1 | 1 | 0 | pitched | SFH | 170,646.9585 | 172,205.8175 |
| 7 | 2268312 | 1946–1970 | 0 | 0 | 0 | pitched | SFH | 172,677.9027 | 174,832.4616 |
| 8 | 2268196 | 1946–1971 | 0 | 0 | 0 | pitched | SFH | 172,612.8326 | 174,867.0901 |
| 9 | 13126791 | 2002 | 0 | 0 | 0 | pitched | Unknown | 173,147.0758 | 172,260.9632 |
| 10 | 2333891 | 1957 | 0 | 0 | 0 | pitched | SFH | 174,227.0597 | 178,078.6134 |

A second source of energy-related data is the energy use (both natural gas and electricity) data provided by Fluvius, the energy network distributor of Belgium. Fluvius provides information of the energy use of buildings, aggregated at the street level or sector level. Building-level data are not available due to privacy issues. The gas use (in kWh) for the year 2017 per m$^2$ building derived from the Fluvius data and assuming an identical use per m$^2$ floor for each building in the street is shown in Figure 2. This very rough estimate does not allow for differentiation of the energy use between the various buildings of one street. Section 5 of this paper explains how this data gap has been handled in this research.

Finally, taking into account information regarding the building owners and occupants will ensure that those with similar renovation interests can be grouped. However, this information is not included in the GIS data of Flanders, nor Leuven. A study conducted by Steunpunt Wonen [15] reports the distribution of the age of the building owner according to the construction year of the building. In this study, a link was also found between the renovation level of buildings and the age of the occupant, and between the houses sold on the market and the occupants' age. Furthermore, the EPC database provides information regarding the property type, distinguishing a natural, social, legal, and governmental owner or landlord. Both the EPC dataset and the results from Steunpunt Wonen are not available at the building level, but these datasets do include statistical data that can be used to enrich the GIS dataset.

A summary of the required data for the goal of this study and their availability for Leuven is provided in Table 5, including the source of the data. The final column provides information about the approach taken to fill the data gaps, further discussed in Section 5.

*4.2. Building Stock Model for Leuven*

The study of the availability of the required data in the previous section reveals that a detailed building-by-building approach for Leuven is not currently possible. Consequently, an archetype approach is chosen. To increase the accuracy of this bottom-up archetype model, GIS data at the building scale are added to the highest extent possible. The stock model developed in this research is, hence, a GIS-enhanced archetype model. The literature review indicated that various building characteristics can be taken into account to define archetypes. In the case of Leuven, the construction period and building type are known for most buildings, so an age–type classification (in order to be able to link the archetypes to the buildings of the stock) seems most appropriate.

These archetypes can then be used to approximate the U values of the building envelope elements in their original state (data gap mentioned before). An important

shortcoming in this approach is that it ignores any renovation that has already taken place, and therefore overestimates the energy use of renovated buildings. Unfortunately, no information is available on the performed renovation measures for the city of Leuven. Section 5 will discuss the methods to fill this data gap.

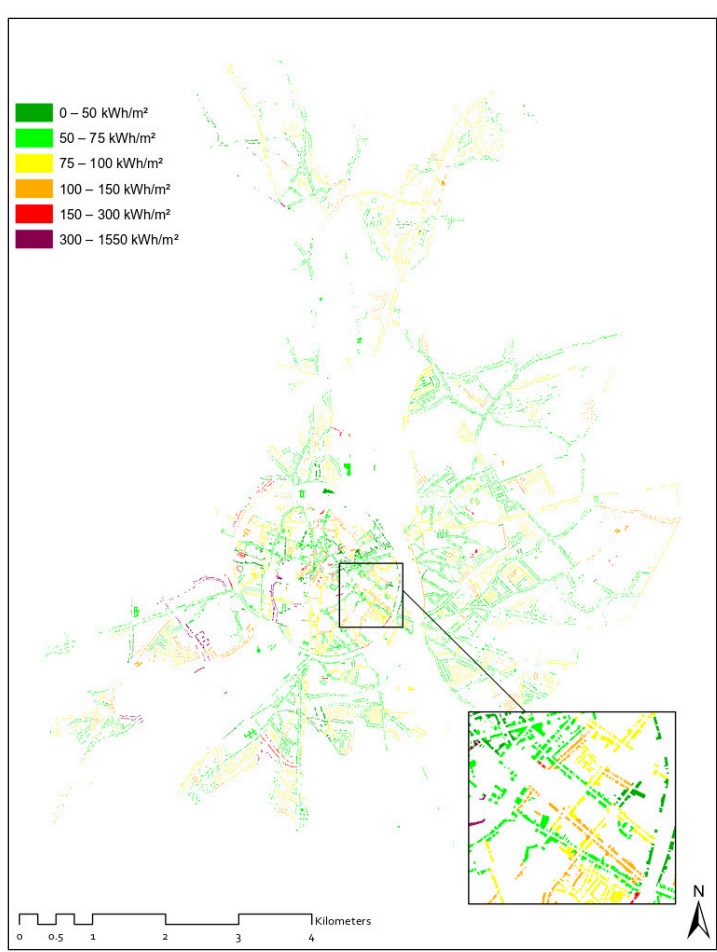

**Figure 2.** Visualisation of the extrapolated gas use per m$^2$ residential building using the Fluvius data at street level (consumption year 2017, in kWh/m$^2$).

**Table 5.** Overview of the data needs, data availability, data gaps, data sources, and methodologies to fill data gaps in Leuven.

| | Data | Necessary for . . . | Availability | Data Source | Methodology to Fill Data Gap |
|---|---|---|---|---|---|
| **Building related data** | Construction year | build-up of the elements | partly | GIS file Leuven | nearest neighbours |
| | Building typology | build-up of the elements, energy loss surface | yes | GIS file Leuven + geoprocessing | - |
| | Geometry (LOD1) | energy loss surface | yes | GIS file Flanders | - |
| | Roof type | build-up of the elements, energy loss surface | partly | GIS file Leuven | statistical distribution based on the building typology and construction year |
| | U value of the building elements | build-up, renovation history | no | | statistical distribution based on EPC based on the building typology and construction year |

**Table 5.** *Cont.*

| | Data | Necessary for ... | Availability | Data Source | Methodology to Fill Data Gap |
|---|---|---|---|---|---|
| | Details heating installation | performance, renovation history | no | | statistical distribution based on EPC based on the building typology and construction year |
| | Construction type (solid/skeleton) | build-up of the elements | no | | statistical distribution based on EPC based on the building typology and construction year |
| | Building address | link with other data sources | yes | GIS file Flanders | |
| | Energy consumption data at street level | average energy performance per street | yes | Fluvius | |
| | Carbon footprint of the city (separately for households and the various energy sources) | top-down validation step | yes | Klimaatactieplan stad Leuven 2020–2025 | |
| **Owner related data** | Rental or property | renovation history, willingness, main goal | no | | statistical distribution based on EPC |
| | Legal status of the owner | renovation history, willingness, main goal | no | | statistical distribution based on EPC |
| | Age of the owner | renovation history, willingness, main goal | no | | statistical distribution based on research Steunpunt Wonen |

## 5. Data Inventory for Stock Model of Leuven

A GIS-enhanced archetype model enriched by energy data was identified as the most suitable model, as not all building parameters are known at the building level. The data needed for the model are shown in Table 5. Figure 3 shows a flowchart, starting with the required data and the available data sources. The archetypes are needed to estimate the original U values and build-up of the building elements. The statistical data are needed to estimate the renovation potential of buildings, which is defined by the building parameters (U values of the existing state) on the one hand, and the building owners on the other. This estimation is made according to the corresponding building typology, function, and construction period. When the location, the original state, and the renovation potential of buildings are known, clusters of similar renovation potential can be identified. The data availability for the case study of Leuven is discussed in Section 4.1, and comparing the available data with the data required reveals that important data gaps remain.

On the one hand, the building parameters for which data are available still have missing values. For Leuven, these are the roof type (1%) and the construction year (40%). On the other hand, some building parameters are not known at the building level for privacy reasons [18,35], and are only known in anonymised datasets, at street level, or in statistical datasets. For Leuven, these are energy- and owner-related building parameters. In Section 3.6, different enrichment methodologies were discussed; the next section discusses various methods to fill the data gaps by enriching the GIS dataset for the case study of Leuven.

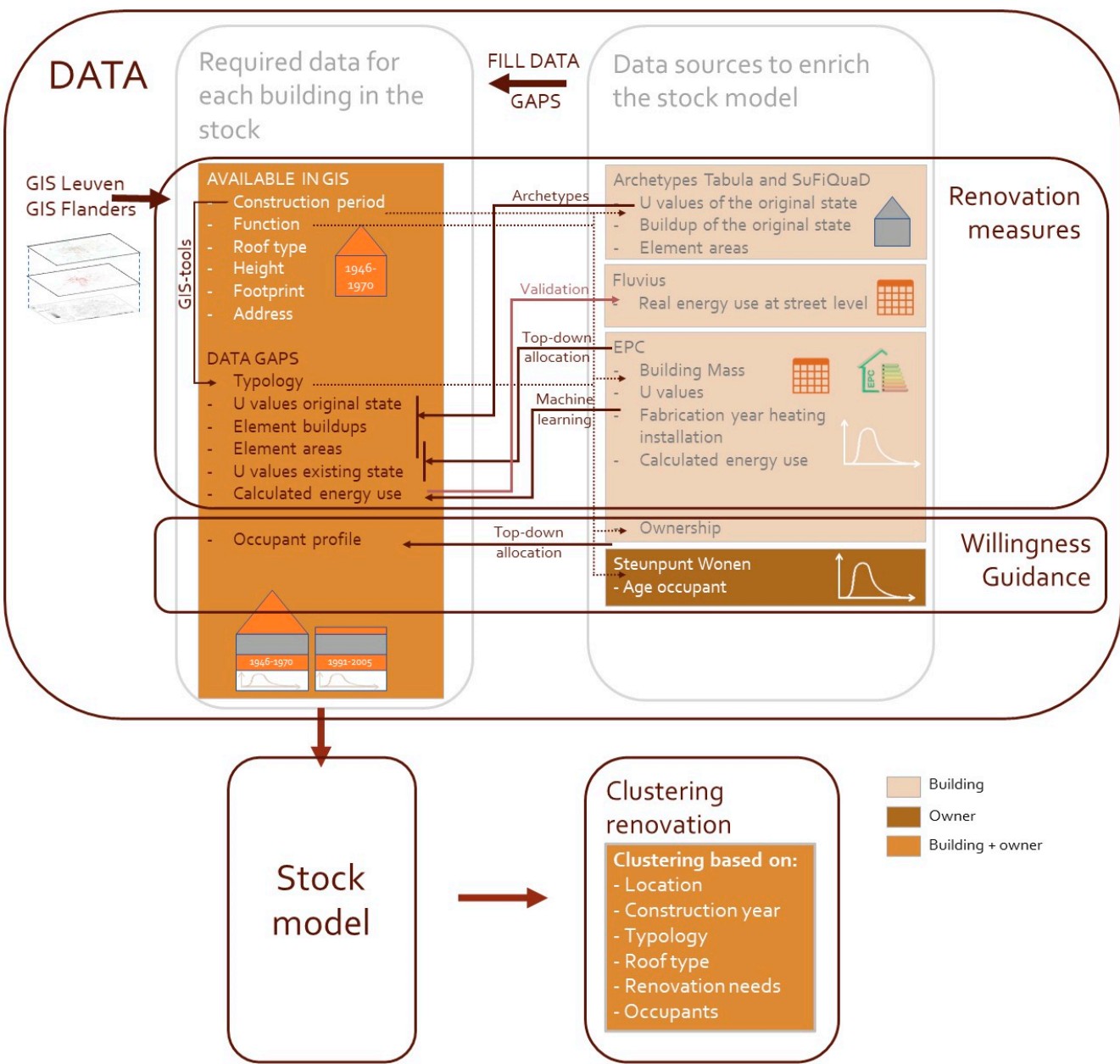

**Figure 3.** Flowchart of data sources for clustering.

### 5.1. Data Enrichment Approaches Used for Modelling Leuven Building Stock

The archetype approach using an age–type classification enables the identification of the original state of the buildings at the moment of construction. In the merged GIS database discussed in Section 4.1, the building typology (e.g., detached house, terraced house, semi-detached house) is unknown. However, the building type could easily be determined using the GIS software Arcmap by counting the adjacent neighbours: detached buildings have no adjacent neighbours, semi-detached houses have one neighbour, terraced buildings have two neighbours, and apartments can be identified by the function (see Table 4). In the GIS database for Leuven, the roof types and construction years were lacking for respectively 1% and 40% of the buildings. These missing values are shown in Figure 4. In an earlier step of this research [61], these data gaps were filled using a GIS enrichment approach similar to the approach used by [40]. The missing roof types were filled in randomly accordingly to the statistical distribution of the roof types of the corresponding building type (construction period and building typology). The data gaps of the construction year could be filled in by taking the average of the construction years of all buildings within a 30 m radius.

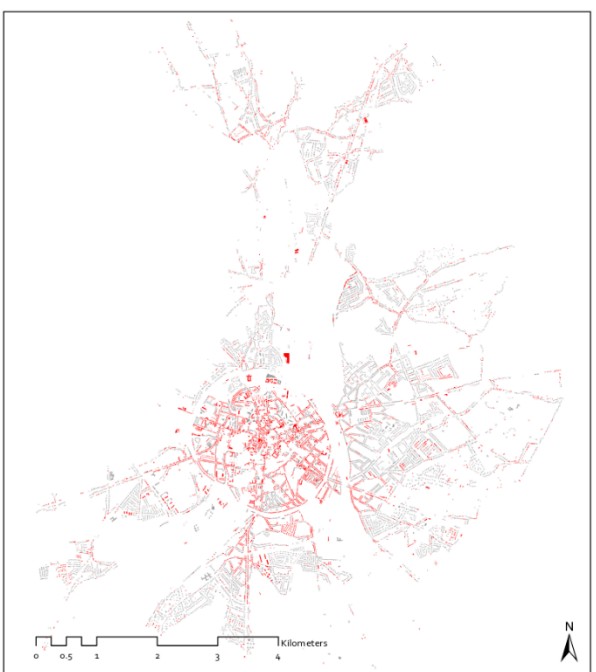 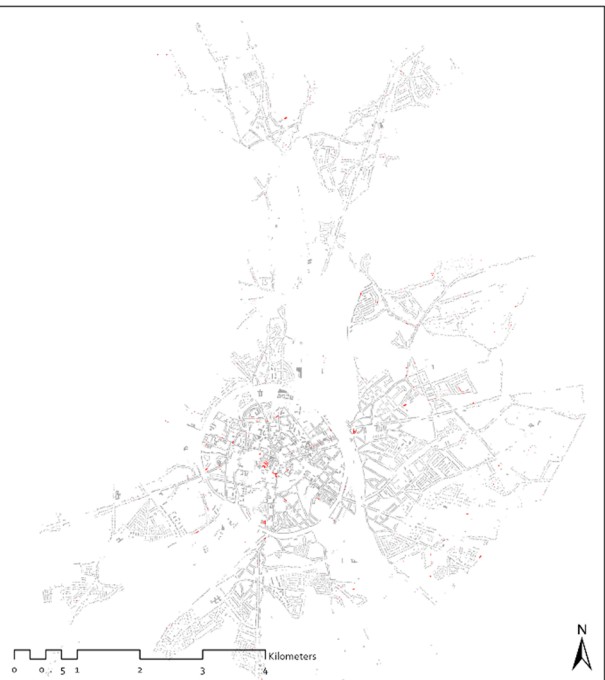

**Figure 4.** Visualisation of the buildings with missing construction years (**left**) and missing roof types (**right**).

To determine the energy use of each building in the stock, machine learning techniques were used, departing from the available building parameters [62]. Different algorithms (decision tree regressor/classifier, random forest classifier, kNN regressor/classifier) were compared, and the kNN classifier (k-Nearest Neighbours) algorithm was identified as the best performing with an accuracy ($R^2$) of 0.89. This methodology is explained in depth in [62].

In order to add energy-related building parameters to the GIS dataset, machine learning techniques were explored in order to determine the U values of the building elements and information on the heating system. However, the U values could not be modelled accurately using these machine learning techniques, as too few EPC database records meet the current standards and, therefore, the EPC database is too unbalanced to predict the U values of building elements. Therefore, all data gaps that could not be resolved using a GIS enrichment algorithm or by a machine learning algorithm were filled in by allocating top-down data randomly to the buildings according to their statistical distribution in the EPC database corresponding to the building type (typology and construction year). Energy parameters are based on the EPC database, while owner-related data are based on the EPC database and the study performed by Steunpunt Wonen [15]. Once the U values have been randomly allocated to the buildings in the stock using the MS Excel function 'Rand' and the corresponding distribution in the stock, these can be compared with the building practice of the corresponding construction period and with current standards. This comparison provides insight into the renovation measures already taken. Based on these comparisons, various renovation statuses can be defined, i.e., 'original' when no renovation has taken place yet, 'small Renovation' when the U value performs better than the original value, but worse than the energy standards, and 'Conform' when the U values are in line with current Energy Performance requirements. Knowledge regarding the original state of the building (based on construction year and build-ups of the archetypes in that period) and the renovation history of buildings, the renovation needs can be defined. In Table 6, the estimations of the U values of the 10 samples of Tables 3 and 4 are shown.

**Table 6.** Ten samples from the EPC database assigned to the GIS dataset of Leuven.

| | | | | Top-Down Allocation | | | |
|---|---|---|---|---|---|---|---|
| NR | Pitched Roof | Flat Roof | Wall | Floor | Window | Year Heating | Ownership |
| 1 | small Ren | small Ren | original | small Ren | Conform | original | Owned–private |
| 2 | small Ren | original | original | original | Conform | small Ren | Rent–social |
| 3 | small Ren | small Ren | original | original | Conform | original | Owned–private |
| 4 | small Ren | original | small Ren | original | Conform | small Ren | Owned–private |
| 5 | small Ren | original | original | small Ren | Conform | original | Rent–private |
| 6 | small Ren | original | small Ren | small Ren | Conform | original | Rent–social |
| 7 | Conform | original | original | original | Conform | small Ren | Owned–private |
| 8 | small Ren | original | original | small Ren | small Ren | small Ren | Owned–legal |
| 9 | small Ren | original | small Ren | small Ren | Conform | small Ren | Owned–private |
| 10 | original | small Ren | original | original | Conform | original | Rent–government |

In conclusion, given the data availability in Leuven, a GIS-enhanced archetype model enriched by (energy) data obtained through GIS additions, a data driven predictive model and top-down allocation, has been chosen as the most suitable approach to enable stock modelling and clustering of buildings. In the next step of the research, this model will be used to cluster buildings based on their renovation needs, estimate the renovation potential of the building stock, and evaluate the potential GHG reductions using an LCA.

*5.2. Uncertainty Assessment for Model of Leuven*

In order to trace back the data reliability, the developed database (stock model) clearly differentiates the measured and estimated (calculated or predicted) data to ensure transparency about the data quality and to allow for the critical interpretation of the results. This also allows the pinpointing of areas where more efforts are needed in future to improve data quality. For Tables 3, 4 and 6, the quality of the data is illustrated in Tables 7–9.

**Table 7.** Ten samples from the dataset for Leuven, including the data quality (SFH: single family house, EDU: education, k: known, n: based on neighbours, u: unknown, g: calculated in GIS, D: detached, T: terraced, SD: semi-detached).

| | | | | | | | | | | | | | | | | GIS | |
|---|---|---|---|---|---|---|---|---|---|---|---|---|---|---|---|---|---|
| NR | ID_Leuven | Construction Year | Quality | Number Floors | Quality | Number Floors Basement | Quality | Number Floors Roof | Quality | Roof Type | Quality | Function | Quality | Typology | Quality | | |
| 1 | 2264224 | 1966 | k | 2 | k | 0 | k | 1 | k | pitched | k | SFH | k | D | g | | |
| 2 | 2264232 | 1971 | k | 2 | k | 1 | k | 0 | k | pitched | k | SFH | k | D | g | | |
| 3 | 2264244 | 1976 | k | 2 | k | 0 | k | 1 | k | flat | k | SFH | k | D | g | | |
| 4 | 2264248 | 1988 | k | 1 | k | 1 | k | 0 | k | pitched | k | SFH | k | D | g | | |
| 5 | 2264252 | 1990 | k | 2 | k | 0 | k | 1 | k | flat | k | EDU | k | D | g | | |
| 6 | 2407336 | 1988 | k | 1 | k | 1 | k | 0 | k | pitched | k | SFH | k | D | g | | |
| 7 | 2268312 | 1946–1970 | n | 2 | k | 0 | k | 0 | k | pitched | k | SFH | k | T | g | | |
| 8 | 2268196 | 1946–1971 | n | 2 | k | 0 | k | 0 | k | pitched | n | SFH | k | T | g | | |
| 9 | 13126791 | 2002 | k | 1 | k | 0 | k | 0 | k | pitched | n | Unknown | u | SD | g | | |
| 10 | 2333891 | 1957 | k | 2 | k | 0 | k | 0 | k | pitched | k | SFH | k | D | g | | |

**Table 8.** Ten samples from the dataset for Flanders, including the data quality (XX: address numbers are anonymised, k: known).

| | | | | | | | | | | | | | | |
|---|---|---|---|---|---|---|---|---|---|---|---|---|---|---|
| **Flanders Dataset** | | | | | | | | | | | | | | |
| **NR** | **ID_Flanders** | **Street** | **Quality** | **Municipality** | **Quality** | **Number** | **Quality** | **Height** | **Quality** | **Perimeter** | **Quality** | **Footprint** | **Quality** |
| 1 | 2880957 | Oude Rondelaan | k | Leuven | k | XX | k | 9.86 | k | 41.07 | k | 104.11 | k |
| 2 | 2880955 | Oude Rondelaan | k | Leuven | k | XX | k | 7.89 | k | 50.28 | k | 154.28 | k |
| 3 | 2880944 | 's Hertogenlaan | k | Leuven | k | XX | k | 6.3 | k | 71.22 | k | 222.35 | k |
| 4 | 2880943 | 's Hertogenlaan | k | Leuven | k | XX | k | 6.1 | k | 54.06 | k | 164.48 | k |
| 5 | 2880937 | 's Hertogenlaan | k | Leuven | k | XX | k | 6.7 | k | 47.7 | k | 130.86 | k |
| 6 | 2885460 | Rotspoelstraat | k | Heverlee | k | XX | k | 7.49 | k | 54.46 | k | 137.34 | k |
| 7 | 2880722 | Brouwersstraat | k | Leuven | k | XX | k | 10.98 | k | 22.58 | k | 31.66 | k |
| 8 | 2880734 | Brouwersstraat | k | Leuven | k | XX | k | 10.93 | k | 38.9 | k | 66.33 | k |
| 9 | 2886057 | Parkbosstraat | k | Heverlee | k | XX | k | 13.5 | k | 54.16 | k | 163.06 | k |
| 10 | 2908196 | Politieke-Gevangenenlaan | k | Wilsele | k | XX | k | 9.01 | k | 40.87 | k | 71.56 | k |

**Table 9.** Ten samples from the EPC database assigned to the GIS dataset for Leuven (m: medium weight, h: heavy weight, O: original, sR: small renovation, C: Conform, r: randomly assigned).

| | | | | | | | | | | | | | | | | | | |
|---|---|---|---|---|---|---|---|---|---|---|---|---|---|---|---|---|---|---|
| **EPC Database** | | | | | | | | | | | | | | | | | | |
| **NR** | **Weight** | **Quality** | **U Roofs Pitched** | **Quality** | **U roofs Flat** | **Quality** | **U Walls** | **Quality** | **U Floors** | **Quality** | **U Windows** | **Quality** | **Heating** | **Quality** | **Occupancy** | **Quality** |
| 1 | m | r | sR | r | sR | r | O | r | sR | r | C | R | O | r | Op | r |
| 2 | m | r | sR | r | O | r | O | r | O | r | C | r | sR | r | Rs | r |
| 3 | m | r | sR | r | sR | r | O | r | O | r | C | r | O | r | Op | r |
| 4 | h | r | sR | r | O | r | sR | r | O | r | C | r | sR | r | Op | r |
| 5 | h | r | sR | r | O | r | O | r | sR | r | C | r | O | r | Rp | r |
| 6 | h | r | sR | r | O | r | sR | r | sR | r | C | r | O | r | Rs | r |
| 7 | m | r | C | r | O | r | O | r | O | r | C | r | sR | r | Op | r |
| 8 | m | r | sR | r | O | r | O | r | sR | r | sR | r | sR | r | Ol | r |
| 9 | m | r | sR | r | O | r | sR | r | sR | r | C | r | sR | r | Op | r |
| 10 | m | r | O | r | sR | r | O | r | O | r | C | r | O | r | Rg | r |

In the next step, the influence of all calculated or predicted data on the end result will be validated by comparing these with the top-down carbon footprint data of the city of Leuven, including all relevant sectors [10]. The most influencing parameters will also be identified and a sensitivity analysis regarding the volatility of the values of the building parameters will be made. This uncertainty will be considered when presenting the results, i.e., a range of results will be shown rather than a single score value. Moreover, this identification of uncertain building parameters with a high impact is of importance to define data collection priorities.

## 6. Conclusions and Discussion

It can be concluded that many different stock models are used in the literature, differing in methodological approach, building classifications, and location specificity. The modelling approach is typically chosen according to the goal and scope of the study, and the data availability. Data availability is a predominant concern in the literature. However, a significant amount of the missing data could be filled in by the energy network operators. Greater data transparency and data access would improve the model's accuracy in forecasting energy use, which can help cities in terms of their sustainability strategies.

These data gaps not only affect the study itself, but the lack of public open data and lack of transparency also have negative consequences on the reproducibility of the study. However, much of the data are still not available because of privacy and confidentiality concerns, but in some countries, more data are becoming available in order to influence energy demand.

The literature review shows that data gaps are either resolved by avoiding the use of these data, or by using various methods such as machine learning algorithms, among others. However, these studies highlight the importance of reliable and accurate data collection to improve the estimation of the current energy use of the building stock and to localise the renovation potential of the stock.

When applied to the case study of Leuven, the lack of building-specific energy data seems the most decisive criterion for the overall approach. This data gap implies that the building stock model of Leuven cannot rely on measured energy data or on thermal building features, which strongly influences the accuracy of the model. Based on the findings of the literature review, a GIS-enhanced archetype model enriched by energy data obtained through a data-driven predictive model is chosen as the most suitable approach. However, completing the dataset using a prediction model instead of measured data has a negative effect on the accuracy of the model. Hence, it is important that assumptions and related reliability issues are displayed in a transparent way to allow for correct interpretations. This implies that far-reaching data management is required to correctly estimate and interpret the results; however, the presentation of this data management is beyond the scope of this paper.

**Author Contributions:** Conceptualization, E.V.; methodology, E.V. and K.A.; software, E.V.; formal analysis, E.V.; data curation, E.V.; writing—original draft preparation, E.V.; writing—review and editing, K.A.; visualization, E.V.; supervision, K.A. All authors have read and agreed to the published version of the manuscript.

**Funding:** This research did not receive any specific grant from funding agencies in the public, commercial, or not-for-profit sectors.

**Data Availability Statement:** The Flanders GIS data that are part of the building stock model of this study are available to download from the Flemish government (https://download.vlaanderen.be/Producten/Detail?id=1&title=GRBgis accessed on 22 February 2019). The GIS data from Leuven that are part of the building stock model are not openly available and were provided by the city of Leuven. However, restrictions apply to the availability of these data, which were used under licence for the current study, and so are not publicly available.

**Acknowledgments:** We would like to thank the city of Leuven for sharing the GIS database.

**Conflicts of Interest:** The authors declare no conflict of interest.

## Abbreviations

ANN: artificial neural networks; BSM: building stock model(ling); EDU: education; EPC: energy performance certificate; GHG: greenhouse gas emissions; GIS: geographic information system; HVAC: heating, ventilation, and air conditioning; LCA: life cycle assessment; LOD: level of detail; SFH: single family house; SVM: support vector machines.

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
