# Peer review of "Developing a Building Stock Model to Enable Clustered Renovation—The City of Leuven as Case Study"

_sustainability, doi:10.3390/su14105769_

Round 1
Reviewer 1 Report
Dear Authors,
Tha paper is interesting however in my opinion you could improve it in some places:
- You have to correct reference numeration, there is some mess. It seems to me also that you have to decide which style you use: [xx] or “Name (yyyy)”
- Lines 39-40: you mention action plan of some cities, please give references to them.
- Lines 59-61: I don’t understand this sentence. Could you explain your thought in more detail?
- Table 1 has no description. There is also now explanation of colors is summary of table (something is in the text, but it should be also I table description).
- Lines 248-250: you mention about paper with top-down approach while in lines 162-163 you decide to focus only on bottom-up models.
- Lines 269 and 288-289: How you distinguish between statistical approach and machine learning approach?
- Lines 288-303: you describe machine learnings research, but only for energy consumption, do you know papers where this approach is also used for prediction of retrofit solution effectiveness?
- Figure 2 is poor quality, could you improv it?
- Figure 3 and Table 5: Pleas propose more informative diagram to presents your database development. As I understand there are some original data sets with some parameters, which are used to find other parameters. E.g. you assign U-value base on some statistic, but is it general statistic or statistic which take into account year of construction which is in one of original datasets? In other words: in ont the Fig. 3 and Tab. 5 there is lack of information on which basis you take randomly value form statistical distribution? You just take randomly value from general statistical distribution (whole building stock) or you have e.g. statistical distributions for various building ages? It seems to me that you are able to prepare diagram which presents this workflow in more detail.
- Line 455: “table 1” is reference which you wanted to point out?
- Lines 460-484: similarly like in p. 7), what is definition of machine learning approach? It seems to me that “first enrichment approach” is also kind of “machine learning algorithm”.
Author Response
REVIEWER 1
We would like to thank the reviewer for his/her valuable feedback to our paper. Your time and efforts for reviewing are highly appreciated.
The structure of the paper has been slightly adapted to combine all findings from literature in section 3 and to group the application to the case study and the recommendations for the case study in sections 4 and 5.
Dear Authors,
Tha paper is interesting however in my opinion you could improve it in some places:
- You have to correct reference numeration, there is some mess. It seems to me also that you have to decide which style you use: [xx] or “Name (yyyy)”
The reference style is discussed with the editor, and in line with other publications in the Sustainability journal. The main reference style is the numeric style in accordance with the journals’ requirements. However, authors and dates can be added to make the subject of the sentence more complete.
- Lines 39-40: you mention action plan of some cities, please give references to them.
References to the action plans of the four cities are added.
- Lines 59-61: I don’t understand this sentence. Could you explain your thought in more detail?
This is now explained in more detail in the text: the renovation measures will be evaluated using an LCA comparison. So the environmental impact of each of the renovation measures will be evaluated. Since multiple material choices and build-ups are possible for each of the identified renovation needs, this will result in a range of results (boxplot). These boxplots allow us to compare the impacts of the different renovation solutions and the boxplots of the different building elements can also be compared to evaluate the impact reduction potentials of the different building elements.
The HVAC-installations and airtightness are not considered for clustering the buildings as the main focus is on renovating the building envelope to investigate the GHG reduction potential by clustered partial or full envelope renovation.
This reasoning is added more clearly in the paper.
- Table 1 has no description. There is also now explanation of colors is summary of table (something is in the text, but it should be also I table description).
The disappeared description is added again.
- Lines 248-250: you mention about paper with top-down approach while in lines 162-163 you decide to focus only on bottom-up models.
Thank you for this observation, the paper with the top-down approach is deleted. - Lines 269 and 288-289: How you distinguish between statistical approach and machine learning approach?
Statistical data, measured data, static/dynamic energy simulations: these are all different data sources for a building stock model. Top-down allocation of statistical data can be used if no other data are available, but have the lowest quality level.
Machine learning is a methodology to fill data gaps when data are lacking for part of the buildings in the stock. Based on the available data of other buildings, predictions are being made for the buildings lacking the same kind of data (e.g. construction year).
The difference between data source and data methodology is now stated more clearly in the paper. - Lines 288-303: you describe machine learnings research, but only for energy consumption, do you know papers where this approach is also used for prediction of retrofit solution effectiveness?
The papers in our literature review were selected based on four criteria:
- the main focus is bottom-up energy modelling or building parameter modelling
- The object of study is a residential building stock, so studies including only a single building or only other building functions are excluded;
- The building stock is located in a western country to provide a good basis for comparison with the case study (Europe, Oceania and the United States of America);
- Projects with multiple publications were only included once.
Therefore papers focusing on effectiveness of renovation measures were not covered in the literature review. In our study the effectiveness of the renovation measures is evaluated using an LCA study, and hence there is no need to predict the effectiveness with machine learning (it will be simulated with energy simulations and environmental impact assessment).
- Figure 2 is poor quality, could you improv it?
The quality is improved.
- Figure 3 and Table 5: Pleas propose more informative diagram to presents your database development. As I understand there are some original data sets with some parameters, which are used to find other parameters. E.g. you assign U-value base on some statistic, but is it general statistic or statistic which take into account year of construction which is in one of original datasets? In other words: in ont the Fig. 3 and Tab. 5 there is lack of information on which basis you take randomly value form statistical distribution? for various You just take randomly value from general statistical distribution (whole building stock) or you have e.g. statistical distributions building ages? It seems to me that you are able to prepare diagram which presents this workflow in more detail.
The statistical distributions are based on the buildings’ typology and construction period. This is added in the text more clearly. - Line 455: “table 1” is reference which you wanted to point out?
This reference is made to the table with all analyzed papers as the methodologies to fill the data gaps all come from the 25 papers studied.
- Lines 460-484: similarly like in p. 7), what is definition of machine learning approach? It seems to me that “first enrichment approach” is also kind of “machine learning algorithm”.
The enrichment of data using GIS is using GIS tools implemented in the GIS software. This is added more clearly in the text.
These GIS-Tools are editing the data, e.g. finding other buildings within a distance, calculating the area of the footprint, summing or averaging a building characteristic for all buildings in one street/of one construction year/…

Reviewer 2 Report
The article "Developing a building stock model to enable clustered renovation - The city of Leuven as case study" explores the literature to overcome the barriers to set and testing a building stock model. I would suggest that authors try to clearly state their objectives since they did not develop precisely a building stock model. To do so, they would need to validate the model outputs.
My understanding is that the authors review other studies' methods to implement them in their case study. Besides clarifying the paper's focus, I do like this study since it explains the current gaps and barriers and strategies to overcome them.
Further recommendations:
- Table instead of table, Section instead of section.
- do not mix author-year references with numbered references.
(ln30) - find another reference to support the figures you present. World energy statistics do not support the 40% energy use and 36% GHG. Are you referring to EU? Final energy? Energy-related emissions?
(ln 57) - clarify that "the goal of this research" is different from the goal of this paper. I think the authors intended to define the research framework or the final goal.
(ln 136)- define western country.
(ln270) - In the context, steady-state is probably more common than static.
(ln395)- Figure 2 quality should be improved. The same applies to Figure 4
(ln428)-Figure 3 and Table 5 should appear after Section 5 tittle.
(ln451)-Section 5.1 instead of 5.2
(ln495)-Section 5.2 instead of 5.3
(ln517)-Why do you need to include Figure 5 results? I think that its inclusion requires a clearer explanation of the applied models, which is not the case.
(ln518)- Explain how did you apply machine learning techniques.
(ln525)- Explain the random process.
Author Response
REVIEWER 2
We would like to thank the reviewer for his/her valuable feedback to our paper. Your time and efforts for reviewing are highly appreciated.
The structure of the paper has been slightly adapted to combine all findings from literature in section 3 and to group the application to the case study and the recommendations for the case study in sections 4 and 5.
- The article "Developing a building stock model to enable clustered renovation - The city of Leuven as case study" explores the literature to overcome the barriers to set and testing a building stock model. I would suggest that authors try to clearly state their objectives since they did not develop precisely a building stock model. To do so, they would need to validate the model outputs.
My understanding is that the authors review other studies' methods to implement them in their case study. Besides clarifying the paper's focus, I do like this study since it explains the current gaps and barriers and strategies to overcome them.
The focus of this paper is stated more clearly in the introduction.
Further recommendations: - Table instead of table, Section instead of section.
This is adapted in the paper. - do not mix author-year references with numbered references.
The reference style is discussed with the editor, and in line with other publications in the Sustainability journal. The main reference style is the numeric style in accordance with the journals’ requirements. However, authors and dates can be added to make the subject of the sentence more complete. - (ln30) - find another reference to support the figures you present. World energy statistics do not support the 40% energy use and 36% GHG. Are you referring to EU? Final energy? Energy-related emissions?
Thank you for your attention. The reference is adapted. - (ln 57) - clarify that "the goal of this research" is different from the goal of this paper. I think the authors intended to define the research framework or the final goal.
The difference between the overall goal of the research and the focus of this paper is stated more clearly in the introduction - (ln 136)- define western country.
By western countries is meant the United States of America, Europe and Oceania. This is added in the paper. - (ln270) - In the context, steady-state is probably more common than static.
Thank you for your suggestion, I changed the energy simulation type to steady-state.
- (ln395)- Figure 2 quality should be improved. The same applies to Figure 4
The quality is improved.
- (ln428)-Figure 3 and Table 5 should appear after Section 5 tittle.
The order is changed.
- (ln451)-Section 5.1 instead of 5.2
Thank you for your attention, I changed the section reference. - (ln495)-Section 5.2 instead of 5.3
The section reference is adapted. - (ln517)-Why do you need to include Figure 5 results? I think that its inclusion requires a clearer explanation of the applied models, which is not the case.
The figure is deleted, and a reference is made to another paper where the models and results are discussed in-depth.
(ln518)- Explain how did you apply machine learning techniques.
A clearer reference is made to another paper where the models and results are discussed in-depth. - (ln525)- Explain the random process.
This random process is implemented using the MS Excel function ‘Rand’ in combination with the distributions in the stock. This is added more clearly in the paper.

Round 2
Reviewer 1 Report
Dear Authors
I regret that you didn't create diagram according to my suggestion but is seems to me that paper coud be accepted in this form.
Kind regards
Author Response
Dear reviewer,
I think there was a small misunderstanding. In the first review round I adapted the figure and table slightly. I now reread your comment and hopefully my new adaptation will meet your expectations.
kind regards,
Evelien
